# Life Cycle Assessment (LCA) of Biochar Production from a Circular Economy Perspective

Joana Carvalho [1,2], Lucas Nascimento [1,2], Margarida Soares [1], Nádia Valério [1,2], André Ribeiro [1,2], Luciana Faria [1], André Silva [1], Nuno Pacheco [1,2], Jorge Araújo [1] and Cândida Vilarinho [2,*]

1   CVR—Centre for Waste Valorisation, University of Minho, 4800042 Guimarães, Portugal
2   MEtRICs—Mechanical Engineering and Resource Sustainability Center, Campus de Azurém, Universidade do Minho, 4800058 Guimarães, Portugal
*   Correspondence: candida@dem.uminho.pt

**Highlights:**

Among several approaches to circular economy and zero-waste concepts, biochar production is a great example and might be a way to push the economy to a carbon-neutral balance. Overall, despite all the differences in assumptions and methodologies adopted, LCA proved that biochar is a very promising way of contributing to carbon-efficient resource circulation, mitigation of climate change, and economic sustainability.

**What are the main findings?**
- Biochar is considered a black porous and carbon-rich matter;
- Biochar is a promising source of alternative energy.

**What is the implication of the main finding?**
- It can be concluded that the costs are closely related to the technologies used in biochar production and also to the feedstock used.

**Abstract:** Climate change and environmental sustainability are among the most prominent issues of today. It is increasingly fundamental and urgent to develop a sustainable economy, capable of change the linear paradigm, actively promoting the efficient use of resources, highlighting product, component and material reuse. Among the many approaches to circular economy and zero-waste concepts, biochar is a great example and might be a way to push the economy to neutralize carbon balance. Biochar is a solid material produced during thermochemical decomposition of biomass in an oxygen-limited environment. Several authors have used life cycle assessment (LCA) method to evaluate the environmental impact of biochar production. Based on these studies, this work intends to critically analyze the LCA of biochar production from different sources using different technologies. Although these studies reveal differences in the contexts and characteristics of production, preventing direct comparison of results, a clear trend appears. It was proven, through combining life cycle assessment and circular economy modelling, that the application of biochar is a very promising way of contributing to carbon-efficient resource circulation, mitigation of climate change, and economic sustainability.

**Keywords:** life cycle assessment (LCA); biochar; biomass; circular economy

## 1. Introduction

As climate change is threatening the world and society grows exponentially, with more and more waste being generated, environmental sustainability is being questioned. It has been proven that linear models promote huge negative impacts on the environment. This occurs by extraction and landfilling at the end-of-life, and also in the economy, since not only when discarded is the economic value of products zero, but the value of finite resources also increases [1,2]. The transition to a circular economy is connected to great

expectations of ecological and economic benefits, helping in the clear separation between economic growth and the use of resources, the building block of the linear economy and its respective impacts, and additionally promoting the perspective of sustainable growth and egalitarian society. The circular models can potentially improve the efficiency of using primary raw materials and allow the waste to return to production as high-quality secondary raw materials [1,3]. In addition, a circular economy can also provide a platform for pioneering methodologies, technologies and business models that create improved economic value from limited natural resources, helping industry to become more resilient to external impacts and improve its global competitiveness.

Among several approaches to circular economy and zero-waste concepts, biochar production is a great example and might be a way to push the economy to a carbon neutral balance [4]. The assembly and appliance of biochar has been widely developed all over the world. Biochar is a solid material formed during the thermochemical decomposition of biomass in an oxygen-limited environment. It is defined, by the International Biochar Initiative, as "a solid material obtained from the carbonization of biomass" [5,6]. Biochar can be produced through several techniques, such as pyrolysis, torrefaction, and gasification. It can also be obtained by various biomass feedstocks, such as wood, agro-residues, or wastewater sludge.

Several studies report the environmental benefits of using biochar in the most diverse industrial areas. However, the safest way to affirm this is through LCA studies that assess the most diverse environmental impacts such as climate change and ozone layer degradation, among others. LCA methodology is a technical tool that allows the systematic analysis and assessment of the environmental aspects and potential impacts associated with products or services throughout its life cycle. In this context, the aim of this work is to perform an overview on biochar concepts and applications and mostly on LCA of biochar production, analyzing several study cases, from the perspective of circular economy.

## 2. Biochar

Biochar is considered a black porous and carbon-rich matter. This material can be produced with little or unavailable air, through a thermochemical conversion of biomass. Chemical, biological and physical properties of biochar make it a great material with many purposes [7]. The following Figure 1 presents some benefits of biochar.

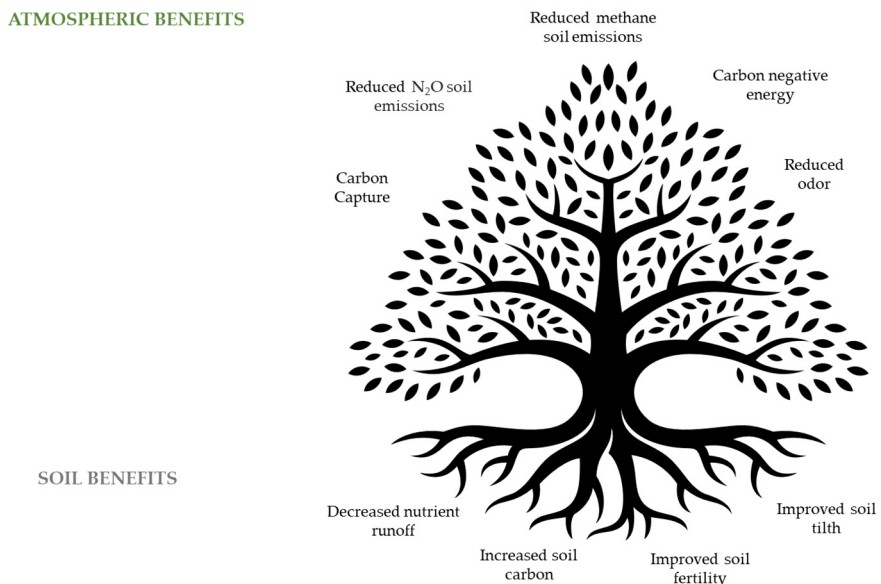

**Figure 1.** Biochar benefits (adapted from [6]).

## 2.1. Biochar Production

For biochar production, several thermochemical processes are used based on different reaction conditions (amount of oxygen available) including pyrolysis, gasification, and torrefaction. In this way, several types of reactors were developed for biomass production, with the objective of achieving the highest quality and yield of the product. These reactors differ in certain thermochemical parameters, such as oxygen availability, temperature, and the rate of heating itself. Thus, these parameters vary the physical and chemical properties of the biochar [8]. The following Figure 2 shows these several methods [8].

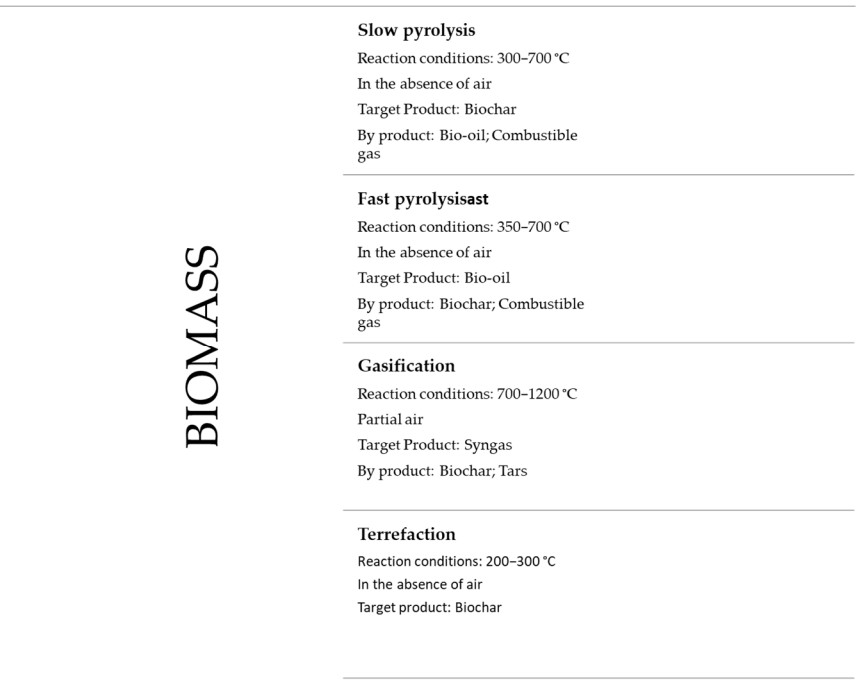

**Figure 2.** Biomass thermochemical conversion technologies for biochar production (adapted from [8]).

Biochar production quality heavily depends on the biomass' compositions and properties. These parameters define the application fields, production, quality and toxicity of biochar. Their stability is influenced by pyrolysis temperature, this being the main parameter [9]. Biochar production involves a complex biochemical reaction process where biomass undertakes decomposition, depolymerization and condensation in anoxic high temperature conditions [4,8,10]. Several factors define the quality of biochar, such as pH, specific surface area, porosity, nutrients, and carbon content, the latter being one of the main factors. High carbon content is directly linked to a high-quality biochar [11].

### 2.1.1. Slow Pyrolysis and Fast Pyrolysis

Different studies have demonstrated that slow pyrolysis allows the production of a quality biochar. This technique is characterized by using moderate temperatures (350–500 °C) and low heating rates [8]. In addition, other operational parameters strongly influence this type of pyrolysis, such as particle size, the atmosphere itself, the use of a catalyst, etc. Veses et al. (2015), stated that improving the residence time of the raw material and the pyrolysis vapor in contact, as well as the particle size and the ratio of catalyst in the biomass can favor the quality and production yield of biochar. Furthermore, it was mentioned that higher pyrolysis temperatures are very important for improving the quality of biochar (expanding carbon content), and reduced heating rate can increase its production in this type of pyrolysis [12].

Unlike the slow process, fast pyrolysis is characterized by the application of high heating rates, temperatures around 500 °C, and residence times shorter than 2 s, with rapid decomposition of the biomass [13]. As mentioned above, higher pyrolysis temperatures

increase the carbon content of the product and its specific surface area [14]. However, Chen et al. (2016), mentioned that increases in pyrolysis temperature, such as an increase in the heating rate, reduces the production yield due to the release of volatile gases. Since there is a shorter residence time, the amount of carbon deposition is reduced too [8].

### 2.1.2. Gasification

Regarding gasification, this process involves an incomplete combustion of biomass using several gasifying agents (air, pure oxygen, steam) that occur at 700–1000 °C [8]. Briefly, the process involves three main reactions: the devolatilization of the biomass, the combustion, and the gasification itself. Char from gasification is different from that obtained through pyrolysis, essentially due to the oxidizing environment of the gasifier. This environment affects the physical, chemical and morphological properties of biochar [15]. As mentioned above, the quality of biochar is directly related to the carbon content present, which is essentially affected by gasification conditions. The main parameters are the equivalence ratio, the properties of the raw material, pressure and the gasifying agent [16]. Several studies have shown that production yield decreases with increasing equivalence ratio, since the gasification temperature increases and carbon content also decreases [17,18].

### 2.1.3. Torrefaction

In turn, the torrefaction process is a modern method of obtaining biochar from biomass. This process is considered a mild pyrolysis, carried out at temperatures between 200–300 °C, under anaerobic conditions [18]. This method uses low heating rates and long residence times [8]. As for fast pyrolysis, increasing the processing temperature leads to a reduction in production yield. However, it is possible to achieve an energetically denser material, correlated with larger destruction of the structural elements. In turn, increasing the process time increases the calorific value of the biomass and has positive effects on carbon content [19,20]. The structure and biomass composition have influence on the process. For example, the size particles and the presence of heavy metals affect the torrefaction mechanism [20].

In short, the higher the carbon content, the more superior the quality of the biochar. In turn, the carbon content is higher when all volatile compounds are released from the biomass. This occurs when slow pyrolysis is carried out, being considered a deeper pyrolysis, using moderate temperatures over a long period of time. In this way, slow pyrolysis yields a high-quality biochar [8]. In torrefaction, the biochar undergoes only a lighter pyrolysis, with a low content of volatile compounds released, and there are not so many chemical reactions [21].

### 2.2. Biochar Applications

The interest in conversion of biomass waste from agriculture and forestry to energy production and carbon sequestration has been growing [22–24]. Carbon and energy content in these wastes make them potential candidates for thermochemical process to produce bioenergy and bioproducts [24]. In general, the many approaches to using biochar facilitate zero waste and the interest of circular economy values [25]. Pyrolysis of biomass waste produces biochar with great capacity in agricultural, industrial and construction applications. In addition, a bio-oil that can be burned to generate hot water, steam and/or electricity, is also produced in the process, thus reducing amount of waste discarded [4]. Recently, the manufacture and use of biochar has been widely developed worldwide. However, there are great uncertainties surrounding the operations, costs and emissions associated with wood processing [26].

As already mentioned, raw material quality, production technology and method conditions mainly define the efficiency, quality and biochar toxicity, and thus affect the subsequent biochar application strategy [4,27,28]. The physicochemical characteristic of biochar reveal its distinct application perspectives, involving soil conditioning, composting additive, building material, activated carbon, promoting anaerobic digestion and a list

of potential uses that continues to expand [5,29–32]. As the number of biochar purposes increases, so does the number of producers. However, it is neither financially nor energetically viable to generate and apply biochar without standards or regulations for its production and application [4]. To solve this problem, many countries are developing their own standards. So far, the International Biochar Initiative (IBI) in the United States of America and the European Biochar Certificate (EBC) are the most-usually used standards all over the world. These two standards were built helping to decrease the health and environmental risks correlated with the production and use of biochar, particularly in agriculture. However, the two standards are voluntary industry guidelines. As biochar has excellent potential for sustainability in several industries, these two guidelines appear to be unsatisfactory to regulate the quality of biochar manufactured as a whole. With the growing importance in biochar, many countries have their individual biochar policies in line with IBI and EBC standards. Other nations, which presently do not have a biochar standard, are controlling biochar usage with fertilizer or compost specifications [4].

Figure 3 shows some examples of promising biochar applications.

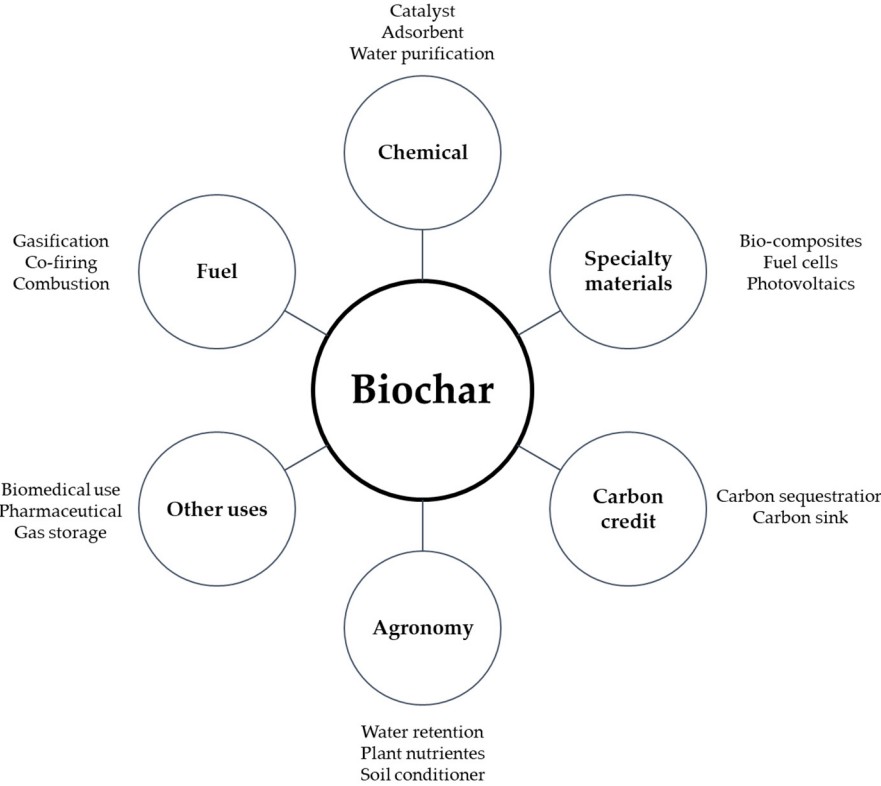

**Figure 3.** Promising applications of biochar (adapted from [33]).

### 2.2.1. Agriculture

Biochar application in agriculture has been extensively tested in the laboratory and field, and it has been usually employed as a chemical component [34–36], or as a soil corrective to enhance crop production, increasing nutrient availability [35,36], water retention capacity [36] and soil microbial activity [37]. Biochar can increase the pH of highly acidic soils [38–40] and mitigate the release of heavy metals into the soil [40–42]. Several studies have shown the advantages of biochar application as a soil conditioner and have demonstrated its potential in that respect. Some of the main benefits are the decreasing of greenhouse gas emissions and the mitigation soil infertility and desertification, improving soil quality and crop yield [43]. This is justified by the inorganic content present in biochar that acts as a nutrient and aids in fertilization [44]. As previously mentioned, biochar is produced by the thermal decomposition of biomass in an oxygen-limited ecosystem while composting is the natural biodegradation of organic material by the microbial community in

aerobic conditions. Carbon-based compounds in compost degrades rapidly and mineralizes so its valuable effects are quite short-lived, which is distinct from biochar, which can endure in soil for a long time [45]. Biochar has also been applied as an additive to enhance the composting rate e.g., a combination of 35% over mushroom compost with 20% biochar has been demonstrated to decrease the composting time to 24 days in comparison to the more common 90–270 days [46]. The elevated porosity of biochar can also enhance microbial development in the compost stockpile and consequently speed up nutrient recovery [47]. The incorporation of biochar in poultry manure for composting condensed the thermophilic stage and increased the highest temperature reached [48]. To manage the quality of biochar, it is extremely essential to determine its organic and inorganic pollutions [49]. General organic and inorganic contaminants that affect biochar quality are polycyclic aromatic hydrocarbons (PAHs), polychlorinated biphenols (PCBs), dioxins, furans and heavy metals [50]. In addition to biochar contaminations, it is also necessary to guarantee that biochar brings more advantages to its applications, such as cation exchange capacity, moisture retention, nutrient retention, plant growth promotion, heavy metals stabilization from soil or construction material, etc. The effectiveness of biochar stabilization of heavy metals can be assigned to the high surface area of the biochar, ability to raise the pH value, small-scale particle, etc. Nevertheless, if biochar is used precisely for its toxicity assessment without soil, a toxic influence is commonly found in living organisms [51]. This negative impact is basically caused by its elevated pH value and potential contaminants, but once biochar is blended with soil in a correct percentage (for example 1% or less), it generally has no noticeable toxic effect and it can improve plant biomass and amount of microorganisms in the soil, which are positive impacts. Studies show that the positive impacts are attributable to the biochar raw material, pyrolysis temperature, biochar percentage and type of soil. The cleaner the biochar raw material is, the less toxic the final biochar product is.

Another biochar application studied is soil remediation and amendment. Biochar has been considered an appropriate matter for this purpose, allowing the absorption and modification of pollutants, through their properties previously mentioned, associated with physical adsorption on the surface and in micropore structures [52].

Biochar has great potential to sequester carbon from materials based on plants, binding $CO_2$ through long-term storage of carbon in soil. This ability is associated with the yield of biochar, the content of stable carbon in biochar and the biochar stability in several circumstances and timeframes of soils [53].

### 2.2.2. Wastewater Treatments

Regarding water purification, biochar can be used in wastewater treatments. Biochar is efficient in the removal of several pollutants, as well as organic compounds (dyes, phenolics, pesticides, polynuclear aromatics and antibiotic), nutrients and heavy metals. Heavy metals are toxic, carcinogenic and non-biodegradable components, so their removal is fundamental [54]. This capacity is justified by their absorbent ability in aqueous solutions due their properties, such as porosity, pH, surface area, surface charge and mineral composition [55,56]. Concerning nutrients (nitrogen, phosphorus, sodium, potassium, etc.), if their concentrations exceed the limits, these components become dangerous to the aquatic ecosystem. Physical and chemical properties associated to biochar are of great significance to the removal of these components, through various removal methods, involving electrostatic interaction, ligand exchange, precipitations with $Mg^{2+}$ or $Ca^{2+}$, complexation with functional hydroxyl, surface sorption, etc. [56–58]. The same is observed in the removal of other organic pollutants, since the efficiency is strongly affected by physical properties, such as specific surface areas and pore-size distribution, surface functional groups and hydrophobic nature [59].

### 2.2.3. Building Materials

Another application of biochar is as a partial component of cementitious building materials, specifically as a sustainable alterative to concrete cement, sand or another energy-

intensive additives used for concrete production [60–63]. Recent studies [61–65] have established that biochar, made from several raw biomass materials under ideal pyrolysis conditions, can lead to an enhancement in strength and biochar-mortar by 15–20% when evaluated to control mixture. The filling of biochar particles can also promote a reduction in capillary water absorption and water infiltration in biochar-mortar mix by 30–40%, thus indicating reduction in water penetration and improved durability of the composite [64]. Heavy metal leaching is a common problem in waste heat treatment products, including ashes and biochar, with a high level of heavy metal content being determined in sources [66]. Heavy metal contained in the biochar can run like leachate when structure builds are exposed to rain and other weathering agents [67].

### 2.2.4. Activated Carbon

Biochar, with a large surface area and pore formation, can potentially be employed as a low-cost activated carbon [68]. Nevertheless, beyond typical pyrolysis treatment of waste biomass, the resulting biochar frequently shows poorer pore behavior than commercial activated carbon. Therefore, several physical and chemical treatment processes were proposed to convert crude biomass or low-grade coal into activated carbon [69]. Normally, physical activation involves elevated temperature in the presence of steam, air, $CO_2$, $N_2$ and inert gases, and chemical activation was carried out utilizing nitric acid, phosphoric acid, sulfuric acid and potassium hydroxide, among others [70–72].

### 2.2.5. Anaerobic Digestion (AD)

AD is an important expertise for organic waste remediation and bioenergy recovery, thus performing a vital role in development the global circular economy [72]. In fact, biochar has been supported as capable of increasing the AD process through quite a few mechanisms. More precisely, it was found that biochar performs a crucial role in increasing AD through several mechanisms, including promoting direct interspecies electron transfer among different microbial species due to biochar's good conductivity [73]; increasing microbial growth through biochar's immobilization effect [74]; adsorbing inhibitory compounds such as heavy metals, ammonia and volatile fatty acids in anaerobic bioreactor [75] and increasing the buffering capacity of the bioreactor due to the relatively high alkalinity of the biochar [75–77].

## 3. Life Cycle Assessment (LCA) Definitions

The concern for the environmental impacts generated by the supply of products and services to society has led to the growth of new tools and methods that aim to understand, control and reduce these impacts. Life cycle assessment (LCA) can be characterized as a compilation and assessment of inputs, outputs and potential environmental impacts of a product or system throughout its life cycle. It is the most widely used assessment type with wide international approval to measure environmental impacts [78,79]. LCA includes entire supply chains, representing all the impacts that occur at different steps and locations throughout the life cycle, regardless of particular processes' physical locations [79].

For an LCA study to be validate, it is necessary to follow the steps stipulated by the ISO 14040 standard, which consists of (i) defining an objective and scope; (ii) carrying out an inventory with as much information as possible about inputs and outputs related to the product or service; (iii) calculating environmental impacts; (iv) systematic and interactive review to validate all information.

Several studies report the environmental benefits of using biochar in the most diverse industrial areas. However, the safest way to affirm this is through LCA studies that assess various environmental impacts, such as climate change and ozone layer degradation among others.

According to Matuski et al. (2020), many authors have used the LCA method to assess biochar projects environmental impact, and these articles are systematically revised to discover a general trend or some pattern. The differences in these studies' contexts and

characteristics do not allow a direct comparison of results, limiting the spectrum of LCA studies. However, biochar application brings substantial benefits, either by neutralizing the emission of greenhouse gases from agricultural production or as a carbon sequestration approach. There is also a great capacity for energy production using synthesis gas and bio-oil byproducts. The advantages of carbon sequestration in biochar and energy production generally overbalance the greenhouse gas emissions produced through the production and handling of the raw material. On the other hand, the effect on another types of environmental impact needs to be assessed and normalized with the intuition of observing some negative effect to guarantee the project's economic sustainability [80].

*3.1. Scope and Objective*

The LCA concept's definition includes the full definition and description of the product, process or activity, the choice of the functional unit to be used and establishes the context in which the valuation must be made, identifying the limits, environmental effects and the methodologies considered for the evaluation.

According to Zhu et al. (2022), for the particular case of LCA of biochar, the objectives and scope are, in general, divided into two fields of research, the first one related to the assessment of the impacts of manufacture and use with association of carbon footprint, midpoint impacts, air pollutants, acidification and other parameters. On the other hand, the second field of research is focused on energy consumption efficiency and economic aspects [81].

Table 1 presents some articles that assess the environmental impacts of biochar production and use. They are divided by a functional unit, reference system, software and impact methodologies used and other focuses.

**Table 1.** LCA case studies for biochar (adapted from [82]).

| Region Country | Functional Unit | Reference System | Allocation + System Boundaries | Ref. |
|---|---|---|---|---|
| Denmark | 1 ton of dry seed | Typical Danish rapeseed to production. | Crop cultivation included. | [83] |
| Belgium, Spain, Italy | 1 kg product 1 ha/yr | Compost, compost blend, mineral fertilizer. | System expansion, cut-off (feedstock), construction and pesticide omitted. | [84] |
| Zambia | Preparation and utilization of 1 kg biochar | x | System expansion—where applicable avoided electricity production (diesel fuel generator wood burning). | [85] |
| Canada | Production of 1 ton (Mg) biochar | Compare two different temperature scenarios. | System expansion, switchgrass cultivation included, energy production offset. | [86] |
| Belgium | 1 ton of biochar | x | System expansion, energy and fertilizer offset, cut-off (manure) þ all (willow cultivation in marginal soils), pyrolysis plant construction included. | [87] |
| Vietnam | 1 ton of rice straw | Open burning of straw, two seasons (spring and summer) modelled, comparison with enriched biochar. | System expansion, open burning of straw eschewed. | [88] |
| Finland | 1 ton oat flakes | Oat flows used as feed or for energy. | System expansion. | [89] |
| China | 1 ton of odt straw | Three straw utilization scenarios: briquetting gasification, pyrolysis two baseline (reference) scenarios: reincorporation, burning | System expansion, offset from avoided fuel consumption. | [90] |

Some authors used LCA study to measure the potential environmental impacts of recovering nutrients from the soil using biochar [84]. Erison et al. (2022), performed an LCA study envisaging the production of biodiesel using biochar as a catalyst [91]. Lefebvre et al. (2021), evaluated the differences between sugarcane biomass and biochar to assess which energy source emitted less $CO_2$-eq in energy supply [92]. The objectives related to the use of LCA in biochar are diversified, and often adapted to particular needs of the research.

The functional unit used varies within each study, e.g., in the study performed by Field et al. (2013), the functional unit was the conversion of 1 ton of dry biomass into biochar. In Hamedani et al.'s (2019), LCA study of biochar, the production of 1 ton of biochar as functional unit was used. Xu et al. (2019), defined the functional unit of their study as a "hectare of agricultural area used for one year" [87,93,94].

When considering the biochar systems' main objective is the use or management of biomass waste streams, upstream functional units are generally used, such as a dry or wet raw material [90]. Another widely applied approach is related to downstream flows, where functional units are primarily specified as the mass of biochar produced or the mass of a crop produced in the treated field. Additionally, it is also common the combination of the two methodologies, for example, a quantity of raw material that produces a certain amount of biochar [95], or even multiple functional units [96]. In the study of Zhu et al. (2022), the author infers that a commonly used functional unit can be the production of 1 kWh of produced energy, stating that, in terms of comparison, one of the most recommended functional units to be used is 1 ton of feed and 1 ton of biochar. [81]

When referring to the boundaries, in the scope of LCA, they can be called "cradle to gate", "gate to gate", "cradle to grave" and "cradle to cradle". Each type of boundary system is described in the literature in accordance with the detail level of study. In brief, the boundary system "cradle to gate" analyses environmental impacts from the extraction of the raw material to the "entrance" at the factory, while the "gate to gate" system measures the environmental impacts of one or more manufacturing processes. The "cradle to grave" expression is used in the assessment of the impacts that cover the extraction of raw materials, the use phase and the final disposal of the product. On its turn, the "cradle-to-cradle" boundary methodology involves recycling and reuse processes [97], as shown in Figure 4.

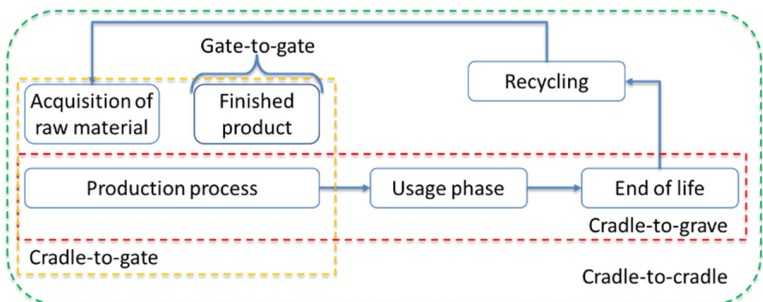

**Figure 4.** LCA boundary system approach.

In a process of biochar production from agro-wastes obtained through pyrolysis, Tiegam et al. (2021), stated that this system boundary can be classified as cradle-to-grave or cradle-to-gate [98]. On the other hand, in the study carried out by Hamedani et al. (2019), involving pyrolysis of pig manure residues for biochar production, its boundary was characterized as cradle-to-grave [87].

In the study performed by Hammond et al. (2011), the frontier used was gate-to-gate with the aim of calculating the impacts of producing 1 kWh of biochar. In another study conducted by Dutta and Raghavam, (2014), the authors infer in their work that the most-indicated boundary system for LCA studies focused on evaluating greenhouse gas was the cradle-to-grave one [99].

### *3.2. Life Cycle Inventory*

Life cycle inventory (LCI) is the collection and quantification of all natural resources consumed and all substances that are emitted into the environment by the life cycle within life cycle phases expressed in the goal and scope definition of this study. The procedure of collecting inventory data begins on the basis of the data requisite for each life cycle stage, in particular the type of data needed for the stage: foreground or background. Foreground data mean that the data are attributable to the target products precisely, while background data can be obtained from special or temporal averages. LCIs are usually based on average data of material and energy inputs (resources and energy used) and outputs (product, waste and emissions produced) collected at a real site or estimated from the literature or from developing studies performed prior to the LCA study [100]. The LCI has a necessary step where elementary flows are included over time and space and is a demanding activity that requires time and attention [101]. It is essential to understand that LCA and LCI are both supported by concepts and assessment of energy consumption and mass balance, based on the first law of thermodynamics related to the principle of energy conservation, by which energy cannot be created or destroyed, but converted into another form. The second law of thermodynamics is also a fundamental concept for understanding LCA, being related to energy degradation; that is, some losses can be due to heat, radiation and others. In this context, LCA makes an energy balance of materials between inputs and outputs and calculates the environmental impacts through the difference between the loss and energy efficiency [102,103].

Some authors declare that the inventory related to biochar production usually focuses on feedstock, transport, pyrolysis process, use of biochar and end-of-life [104]. Dutta and Raghavam (2014), in their work consider inventory, obtaining feedstock analysis, all transports associated with biochar, pyrolysis process to obtain biochar and use of biochar for energy production [99]. In the study carried out by Llorach-Massana et al. (2017), the inventory was focused on the feedstock, where the feasibility of using tomato plant residues from an urban garden was studied [105]. In other studies, the benefits of producing biochar from rice field residues were analyzed [88].

In the study performed by Matuštík et al. (2020), LCI for biochar must contain information on the raw material, land use, transport, energy used in pyrolysis, pyrolysis gas emissions and, depending on the functional unit and borders, waste must be also accounted for [82].

### *3.3. Life Cycle Impact Assessment (LCIA)*

To understand the potential environmental impact of the use of natural resources and the attempts created by the system, the LCIA stage is essential. In this methodology, flows resulting from a product's life cycle are aggregated into impact categories [82].

The same studies presented in Table 1 were compiled in Table 2 with LCIA methodologies, impact assessment and GWP results.

According to de Vries and de Boer, (2010), the environmental impact types considered during the life cycle of a product are related to the use of resources such as earth or fossil fuels and the emission of the pollutant's ammonia or methane. Ferrão, (2009), and Santero et al. (2011), considered categories of impacts on climate change, eutrophication, acidification, depletion of resources and ecotoxicity, among others, where each type of emission is related to one or more impacts [102,106,107]. The impact assessment uses the inventory results to evaluate the potential of environmental impacts while providing information for interpretation. In this phase, statistical techniques, such as weighting, normalization and aggregation, are essential to comparing impact types and measuring their relevance [100].

**Table 2.** Study cases with LCIA methodologies, impact categories analyzed and GWP results (adapted from [80]).

| Study | LCIA Methodology | Impact Assessment | Results | Ref. |
|---|---|---|---|---|
| Thers et al. (2019) | IPCC (2013) | GWP 100 yr, 20 yr | 171 kg $CO_2$-eq/Mg dry seed (400 °C) and 111 kg $CO_2$-eq/Mg dry seed (800 °C) compared to 638 kg $CO_2$-eq/Mg dry seed for oilseed rape cultivation without biochar amendment = significant GHG reduction. | [83] |
| Oldfield et al. (2018) | CML | GWP, AP, EP | Net negative—single value not presented. | [84] |
| Smebye et al. (2017) | ReCiPe 2016 endpoint | GWP, PMF, land occupation, FDP, þ all ReCiPe 1 endpoint categories | Only endpoint GWP value calculated. | [85] |
| Brassard et al. (2018) | IPCC 2007 | GWP | Scenario A (459 °C) −2110 kg $CO_2$-eq/Mg biochar, Scenario B (591 °C) −2561 kg $CO_2$-eq/Mg biochar. | [86] |
| Hamedani et al. (2019) | IMPACT 2002+, CML | All midpoint (IMPACT) and endpoint (CML) | IMPACT 2002+: −2063 kg $CO_2$-eq/t biochar (willow) and −472 kg $CO_2$-eq/t biochar (pig manure); CML: −2089.65 kg $CO_2$-eq/t biochar (willow) and −466 kg $CO_2$-eq/t biochar (pig manure). | [87] |
| Uusitalo and Leino, (2019) | IPCC (2013) | GWP | Biochar production from oat side-flows leads to GHG emission reduction of 350 k g$CO_2$-eq/t oat flake, buffer zone biomass biochar potential of 390 kg $CO_2$-eq/t oat flake. | [89] |
| Clare et al. (2015) | IPCC 2007 | GWP | −1.35 Mg $CO_2$ e/odt straw | [90] |

There are several methodologies for calculating environmental impacts such as Recipe, CML, Usertox, Carbon Footprint IPCC for parameters and others. According to Zhu et al. (2022), the choice of impact category must converge with the goals. Yang et al. (2018), used the CML methodology; Thers et al. (2019), used the IPCC methodology; Oldefild et al. (2018), applied CML methodology and Smebye et al. (2018), assessed the impact through ReCiPe methodology [82–85,103].

Several authors studying LCA on biochar have used methods such as Eco Indicator 99 and the ReCiPe midpoint approach to evaluate the entire biochar chain [108]. In the study carried out by Hamedani et al. (2019), the use of two methodologies, IMPACT 2002+ and CML, was applied to calculate environmental impacts [87].

When looking at Table 2 and considering all the differences in the limits of the system such as functional units, among other parameters, the LCA results are not comparable. The GWP results presented make a general balance of GHG that varies in value according to the methodology used. According to a study by Brassard et al. (2018), in their work they used biochar for soil correction and observed a GWP impact of up to 2561 mg $CO_2$-eq/t of biochar [86]; Mohammadi et al. (2016), revealed the carbon footprint of rice production with a change in the biochar of 3.85 kg $CO_2$-eq/kg. Nevertheless, the results showed similar tendencies. However, when looking closely, the results show a certain tendency. It is remarkable that the biochar–soil correction systems show a clear advantage from the point of view of climate change [88]. Biochar production appraised the neutralization's impact of agricultural production when GHG emissions were positive, such as in the case presented by Thers et al. (2019), and other studies. The processing of raw materials and especially the pyrolysis system are the most important causes of GHG emissions, not including agricultural production's impact. In relation to the impact, it is normally considered residual. The benefit of carbon capture and energy production from biochar,

in addition to the advantages associated with the co-products generated, compensate the GHG emissions caused by the biochar production itself [83].

Biochar application can contribute to carbon sequestration by increasing production, reducing the use of fertilizers and reducing $CH_4$ or $N_2O$ emissions, among other factors [84].

Other studies have been carried out with the aim of evaluating the influence of biochar production on the impact categories. Concerning acidification and eutrophication potential, agricultural processes, such as the application of fertilizers, are the most important sources of negative impacts [84], as well as with regard to the impact of ecotoxicity categories [87]. Electricity from the network for pyrolysis and agricultural operation was reported as another source of negative impact on acidification and eutrophication potential [109].

## 4. Life Cycle Cost Assessment (LCCA)

For a process to be viable, in addition to the environmental aspects, it is mandatory to assess their associated costs. Life cycle cost assessment (LCCA) is a method of assessing the total cost of a process. It takes into account all costs of acquiring, owning, operating and disposing. LCCA is especially useful when project alternatives that fulfill the same performance requirements differ with respect to initial costs and operating costs and should be compared in order to select the one that maximizes net savings best.

In this context, conducting a complete LCCA of biochar-based products is an essential step in the breakeven and viability of this technology during the project or industrial application period.

Durairaj et al. (2002), reviewed models for LCCA [110], namely: (i) LCCA model of Fabrycky and Blanchard; (ii) LCCA model of Woodward [111]; (iii) LCCA model of Dahlen and Bolmsjo [112]; (iv) the activity-based costing (ABC) model [113]; (v) the economic input–output LCA model [114]; (vi) the design-to-cost model (DTC) [115]; (vii) the product life cycle costing (PLCC) to manufacturing system [116]; (viii) the total cost assessment (TCA) model.

Fabrycky and Blanchard presented an elaborated LCCA model to approach detailed cost analysis of all the costs related to the entire life cycle of either product.

The major advantage of this model is found in its detailed cost breakdown structure (CBS). First, they split the total cost of a product or a system into four categories, respectively: (1) research and development costs; (2) production and construction costs; (3) operation and maintenance costs; (4) retirement and disposal cost. This model comprises the essential features of a holistic methodology to assess the life cycle and determine the total cost of a product.

Woodward's LCCA model was made to focus on planning and monitoring assets over their entire life cycle—from the development stage through to disposal. The minimum life cycle cost of the asset is given by the optimization of the trade-off among the cost factors. This process requires the estimation of whole-life costs prior to making a purchase choice for an asset from the accessible alternatives. This approach emboldens a long-term perspective on the investment decision-making procedure [111].

Dahlen and Bolmsjö's LCCA model aims to extend the application's field and perform an investment analysis when raising the production factor—labor. It considers all the costs associated with an employee over their employment time. The costs of labor can be graphed like the costs over the life cycle for production equipment [112]. In turn, the ABC Model has the most likely-looking cost-effective evaluation in lifecycle design. Respecting environmental matters, uncertainty must be considered due to the predominant lack of hard data. When there is a lack of information and the existence of unexpected activities, uncertainty conditions must be also used [113]. The economic input–output LCA model is a new tool that can complement the conventional LCA and overcome its limitations. The objective is to augment conventional economic input–output tables with appropriate sectored environmental impacts indices, which are then used to analyze economywide, direct and indirect environmental impacts of changes in the output of selected industries.

This model represents all the supplier relationships in the supply chain for industrial production. Its application was shown to be feasible, rapid and inexpensive [114].

In relation to the DTC model for manufacture systems, it provides a common methodology to merge cost modelling and quality function deployment (QFD) to assess the possible trade-offs among the costs and performance of product alternatives in the initiation of the production system's design. The design-to-cost approach has a plan for choosing a system design [115].

The PLCC model determines the life cycle expenses of capital goods such as machines and manufacturing systems. In this methodology, single processes connected to the product's life cycle are expressed. With an intent to the redesign of present product structures, it is feasible to derive approaches out the cost structures of the life cycle. The early stages of the product life cycle are production, use and disposal or de-production. The design of the product should be directed upon the needs of the use phase to reduce costs in the several phases of the life cycle. A similar dependency comes up in the disposal phase [116].

The TCA model seems to be the most helpful and practical tool for small manufacturers. It supplies a streamlined approach to identifying and quantifying the costs of pollution prevention investments. It expands the scope of capital budgeting to comprise indirect benefits, thereby increasing the magnitude of savings originated from pollution prevention investments. The information requirements of TCA can be readily implemented in the small business scenario.

It is not possible to create a single LCCA model that considers all the requirements, yet it is possible to develop models to answer specific needs, such as the elaboration of an ecological, sustainable and economical product.

Most of the related studies in the literature focus on the economic assessment of biochar systems, such as the studies carried out by Roberts et al. (2010), and Galinato et al. (2011). In these studies, is typical to find that the potential profitability of biochar production systems is highly dependent on the feedstock used. In their study, Yoder et al. (2011), proceed to model the trade-off between product yield and product quality as the conversion temperature increases, exploring the implications of different production techniques and resulting variations in the properties of biochar for overall system performance [117–119].

In another study, Homagain et al. (2016), found that, within the limit of life cycle analysis, the economic viability of the biochar-based bioenergy production system is directly dependent on pyrolysis costs and raw material processing (drying, grinding and pelletizing), in on-site collection, and also in total carbon offset amount provided by the system. Through a sensitivity analysis of the transport distance and the displacement values, it was shown that the system is profitable in the case of high biomass availability within 200 km and when the cost of carbon sequestration exceeds the 60 Canadian dollars-per-ton carbon equivalent ($CO_2$-eq) [120].

In the study performed by Clare et al. (2015), it was found that straw briquetting for thermal energy is the most economical carbon reduction technology, requiring a subsidized CAD 7 mg$CO_2$-eq. However, China's current bioelectricity subsidy scheme makes gasification (net present value (NPV) CAD 12.6 million) more financially attractive to investors when compared to briquetting (NPV CAD 7.34 million) and pyrolysis (NPV CAD 1.84 million). The potential for direct carbon reduction from pyrolysis (1.06 mg$CO_2$-eq per odt straw) is also less than briquetting (1.35 mg$CO_2$-eq per odt straw) and gasification (1.16 mg$CO_2$-eq per odt straw). The authors conclude that the indirect carbon reduction processes that results from biochar utilization can significantly improve the pyrolysis scenario and carbon reduction potential, bearing in mind that improving the agronomic benefit of biochar is essential for the pyrolysis scenario to compete as an economically viable and cost-effective mitigation technology [90].

According to the study performed by Cleary, (2018), the added value of pyrolysis for biochar production is more profitable than selling highest quality wood chips for cellulose. The modelled biochar price ranged from CAD 3 to CAD 4/kg, quoted for 10 to 20 kg of biochar packages. The pyrolysis cost was estimated at about CAD 150,000.00, the operating

cost around CAD 78,840.00, including labor and electricity. Thus, it can be concluded that the costs are closely related to the technologies used in biochar production and also to the feedstock used [121].

**5. Final Remarks**

Two of the most important domains facing specific challenges within the circular economy are biomass and bio-based products. Materials based on biological resources can be used for a wide range of products and energy uses. The bioeconomy offers alternatives to products and energies based on fossil fuels and can contribute to the circular economy. Bio-based materials can also have advantages linked to their renewability, biodegradability, or the possibility of composting. On the other hand, the use of biological resources requires attention to be paid to the environmental impacts of its life cycle and to its sustainable supply. From a circular economy perspective, the cascading use of renewable resources should be promoted, where appropriate, with various cycles of reuse and recycling. Bio-based materials can be used in multiple ways, with the possibility of reusing and recycling them several times, which is consistent with the application of the waste hierarchy and, more generally, with options that lead to the best overall result for the environment.

Biochar is a solid, carbon-rich material generally obtained from thermochemical conversion of biomass and respective carbonization in oxygen-limited environments and has been proposed as a potential solution to climate change, energy security, degradation of natural resources, food security and catastrophic forest fires worldwide. Biochar production implies a complex chemical reaction process where biomass undergoes decomposition, depolymerization and condensation in anoxic high temperature conditions. As was extensively discussed, the many strategies for using biochar facilitate zero waste and the development of the circular economy. In addition, LCA has proven an excellent tool for quantifying the potential of biochar utilization, as well as for fostering and managing its production. Nevertheless, as was shown, multi-purpose applications of biochar make the functional units and system boundaries of different cases variable. Additionally, although these variables can be contextualized, they can barely be eliminated. Therefore, extensive system boundaries and more inclusive inventory considerations must be integrated and comprehensively analyzed in the LCA of biochar production. Although solid international guidelines and frameworks are available to promote consistency, and most of the studies follow the ISO standard for LCA, they still may consider different criteria and assumptions, resulting in different outcomes, so the methodology should be unified allowing to compare the results, at least to some extent. In addition, economic analysis through LCCA should be encouraged to optimize the flows of sustainable biochar production.

Overall, despite all the differences in assumptions and methodologies adopted, LCA proves that biochar is a very promising way of contributing to carbon-efficient resource circulation, mitigation of climate change and economic sustainability.

**Author Contributions:** Conceptualization J.C. and L.N.; writing—original draft preparation, J.C., L.N., M.S. and N.V.; writing—review and editing, A.R., L.F., A.S. and N.P.; project administration, J.C. and J.A.; funding acquisition, C.V. All authors have read and agreed to the published version of the manuscript.

**Funding:** This work was co-financed by Compete 2020, Portugal 2020 and the European Union through the European Regional Development Fund—FEDER within the scope of the project WAST'AWARENESS—Technology Transfer in Waste Valorization and Sustainability (POCI-01-0246-FEDER-181304).

**Conflicts of Interest:** The authors declare no conflict of interest.

**Abbreviation**

AD       aerobic digestion
EBC     European Biochar Certificate
GHG     greenhouse gas
GWP     global warming potential
IBI       International Biochar Certificate
LCA     life cycle assessment
LCI      life cycle inventory
LCIA    life cycle impact assessment
PAHs    polycycle aromatic hydrocarbons
PCBs    polychlorinated biphenyls

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
