# Peer review of "Life Cycle Assessment (LCA) of Biochar Production from a Circular Economy Perspective"

_processes, doi:10.3390/pr10122684_

Round 1

Reviewer 1 Report

The manuscript entitled “Life Cycle Assessment (LCA) of biochar production in a circular economy perspective” presents a review article covering extensive research life cycle assessment on biochar production and application in a circular economy perspective. The review article is well-written and covering gate to gate assessment. However, some minor mistakes need to be corrected before acceptance. The following are the main points to be improved:

Comments and Suggestions for Authors

1.     The biomass acquisition and choice of biochar production are missing from the LCA perspective.

2.     Inconsistency in writing format, formula writing. Please carefully write the chemical formula (subscript writing).

3.     Very little information about methodologies used to carry out LCA and LCCA.

4.     This study was based on carbon neutrality, but a negative carbon emission perspective is missing.

5.     The reference style may need to be corrected.

6.     The abbreviation list and highlights are missing and must be provided.

7.     The NPV presented is a general value irrespective of the plant capacity.

8.     The cost assessment for biochar is not clear.

Reviewer 2 Report

The paper addresses the life cycle assessment for biochar production with a circular economy perspective.

The manuscript considers a suitable research gap in terms of life cycle assessment but the circular economy perspective is unclear.

Added value of the paper: An analysis of different case studies are done but they should be compared more critically. How are the case studies adapted from ref 81?

The paper considers too many aspects but the level of detail for them is not enough. The cost aspect is not preoperly justified as it is based on a broad estimation. How is this reliable?

The paper lacks of scientific critical discussion when comparing the case studies. The table 1 is not fully analysed. Why do they use different basis unit? If converted to the same, is it the obtained value similar?

Conclusions are too broad and they do not add particular values to the main study.

Round 2

Reviewer 2 Report

Paper has improved. It should be published.